# The Supplementation of Bee Bread Methanolic Extract to Egg Yolk or Soybean Lecithin Extenders Can Improve the Quality of Cryopreserved Ram Semen

**DOI:** 10.3390/cells11213403

**Published:** 2022-10-27

**Authors:** Asmaa E. Sharaf, Wael A. Khalil, Ezz I. Khalifa, Mohamed A. Nassan, Ayman A. Swelum, Mostafa A. El-Harairy

**Affiliations:** 1Department of Animal Production, Faculty of Agriculture, Mansoura University, Mansoura 35516, Egypt; 2Animal Production Research Institute, Agricultural Research Center, Ministry of Agriculture, Dokki, Giza 12618, Egypt; 3Department of Clinical Laboratory Sciences, Turabah University College, Taif University, P.O. Box 11099, Taif 21944, Saudi Arabia; 4Department of Theriogenology, Faculty of Veterinary Medicine, Zagazig University, Zagazig 44519, Egypt

**Keywords:** ram, semen, cryopreservation, extender types, bee bread extract, sperm characteristics, antioxidant, apoptosis

## Abstract

Bee bread has numerous nutritional benefits and bioactive compounds. Other bee byproducts have been used as extender additives to improve semen cryopreservation. Here, we examined the effects of supplementing egg yolk extender (EYE) or soybean lecithin extender (SBLE) with bee bread extract (BBE) on the quality of cryopreserved ram semen. Semen was collected from five adult Rahmani rams once a week for 7 weeks. EYE and SBLE were supplemented with BBE. Antioxidant capacity and total phenolic compound, total flavonoid compound, and total soluble carbohydrate levels of BBE were measured. Sperm characteristics, including progressive motility, viability, abnormalities, membrane integrity, and acrosome integrity, were analyzed after equilibration, thawing, and thawing followed by a 2-h incubation. The total antioxidant capacity and malondialdehyde, hydrogen peroxide, aspartate transaminase, alanine transaminase, alkaline phosphatase, and total acid phosphatase levels in extenders were determined after thawing. Sperm apoptosis was analyzed using annexin V assays. SBLE was more effective than EYE for cryopreserving ram semen. Extender supplementation with BBE improved ram semen quality during freezing in a concentration-dependent pattern. Motility, vitality, and membrane integrity were particularly enhanced in BBE-treated semen. Additionally, BBE promoted antioxidant and enzymatic activities and reduced apoptosis in semen. Thus, extender supplementation with BBE improved sperm cryopreservation.

## 1. Introduction

Sheep rearing is gaining popularity in many countries, and farmers aim to increase sheep reproduction by selecting the best breeders. Artificial insemination is an efficient tool using cold and frozen–thawed extended semen from stud animals and is key for genetic improvement programs. The sperm are particularly susceptible to cold shock, and the successful cryopreservation of semen depends on dilution–cooling–freezing and thawing protocols [1]. Oxidative stress is characterized by excessive production of reactive oxygen species (ROS) and leads to the oxidation of various cell components [2]. To reduce the adverse effects of oxidative stress on ram semen, semen extender supplements are used [3,4].

Egg yolk is used in most semen extenders to protect sperm from cold shock. Additionally, it contains antioxidants that protect the walls of sperm cells from oxidation. Adding Tris-egg yolk to semen extenders has been effective in improving semen characteristics compared with those obtained with other extenders [5]. However, egg yolk might have some adverse effects including bacterial contamination and transmission of diseases [6,7]. Egg yolk may also affect the fertility of semen and has been replaced by soybean lecithin, which is effective in reducing the damage caused by cold shock and preserving the plasma membrane [8]. Thus, soy lecithin is a plant-based alternative used in semen extenders to preserve sperm cells from cold shock during cryopreservation [9].

Many authors have studied the effects of various natural compounds as additives to semen extenders for promoting the quality of the extenders and, consequently, sperm characteristics [10,11,12,13,14] after cooling and thawing. These natural additives comprise bee byproduct extracts such as propolis [15,16,17], royal jelly [18,19], and pollen grains [20,21,22]. Studies have attributed the enhancement of sperm activity by bee byproduct extracts mainly to the extract’s rich content of nutrients [23] such as sugars, minerals, and vitamins (B1, B2, C, B6, B5, and B3). Bee bread (BB) is a bee byproduct. It results from the lactic fermentation of pollen grains that were collected by bees from flowers of melliferous plants, mixed with bees’ digestive enzymes, carried into a hive, and kept with a thin layer of honey and bee wax [24]. Mohammad, et al. [25] found that BB contains proteins, lipids, carbohydrates, ash, moisture, energy compounds such as fructose, glucose, sucrose and maltose, vitamin C, essential amino acids, minerals, and heavy metals. Hashem, et al. [26] confirmed that BB is enriched in components including monosaccharides, disaccharides, fatty acids, vitamins, minerals, enzymes, and phenolic compounds. They also showed that the BB has potent antioxidant and antibacterial properties as flavonoids possess free radical-scavenging activity and thereby prevent free radical-induced DNA damage.

However, researchers have focused mainly on adding propolis, royal jelly, and pollen grains to semen extenders, and there are no previous reports dealing with BB extracts (BBE) usage for semen extender preservation and storage. Therefore, the present study aimed to investigate the effects of supplementing soybean lecithin or egg yolk-based extenders with different concentrations of BB on sperm characteristics, oxidative stress, enzymatic activity, and apoptosis of frozen ram semen.

## 2. Materials and Methods

### 2.1. BB Extraction Procedure

BB was collected at the end of the winter from colonies of honeybees located in the Dakahlia governorate, Egypt. It was kept in the dark at −20 °C until its processing. The extraction of BB was carried out using methanol according to the method published by Markiewicz-Zukowska, et al. [27] with the following modifications: BB was ground into a fine powder and 50 g was mixed with 200 mL of methanol (95%). The mixture was soaked for 48 h and then filtered using Whatman paper No. 1. The filtrate of BBE was evaporated (40 °C) in a rotary evaporator (Chemker410, Rocker, Kaohsiung City, Taiwan) and then kept at −20 °C until use. One gram of extract was dissolved in 4 mL of DMSO to facilitate its addition to the medium.

### 2.2. BBE Analysis

The antioxidant activities of BBE were measured using the 2,′-diphenyl-1-picrylhydrazyl (DPPH) scavenging assay [28,29]. The ferric reducing ability (FRAP) assay was conducted according to Benzie and Strain [30] with minor modifications, allowing us to perform the measurements using microplates. The levels of total phenolic and flavonoid content (TPC and TFC, respectively) in the BBE were determined using a FluoStar Omega microplate reader (BMG Labtech, Ortenberg, Germany) as described by Attard [31] and Kiranmai, et al. [32], respectively. A total soluble carbohydrate (TSC) assay was performed following the method of Gerhardt, et al. [33]. Detailed methods are included in the Appendix A.

### 2.3. Animals, Semen Collection, and Initial Evaluation

Five mature Rahmani rams were used as semen donors. They were housed individually in barns. The semen quality was tested and acceptable fertility rates were found. The rams had no health problems, were aged 33–36 months, and had an average live body weight of 75.0–80.5 kg. They were kept at the El Serw Station, Animal Production Research Institute (APRI), Agriculture Research Center, Ministry of Agriculture, Egypt. All rams used in the experiments were fed with concentration to roughage ratios (60% concentration to 40% roughage) meeting the recommended allowances for different physiological and production phases. Mineral blocks and fresh water were available ad libitum throughout the experimental period. All semen analyses were conducted at the Physiology and Biotechnology Laboratory, Animal Production Department, Faculty of Agriculture, Mansoura University, Egypt. The study lasted from September 2021 to January 2022. Semen ejaculates were collected from each ram using an artificial vagina once a week for seven weeks. Semen samples were initially evaluated for volume, progressive motility, viability, abnormality, and sperm concentration. Semen samples of volume ≥0.8 mL, progressive motility ≥75%, live sperm ≥85%, abnormal sperm ≤15%, and sperm concentration ≥2.9 × 10^9^/mL were pooled, extended, and used for experiments. The study was approved by the Scientific Research Ethics Committee of Mansoura University following the Animal Research Reporting of In Vivo Experiments guidelines.

### 2.4. Preparation of Semen Extenders and Freezing–Thawing Procedure

Two basic semen extenders, egg yolk extender (EYE) and soybean lecithin extender (SBLE), were prepared. They contained 3.025 g Tris (Sigma Chemical Co., St. Louis, MO, USA), 1.66 g citric acid monohydrate (Sigma), 1.25 g glucose (Sigma, Aldrich), 20% egg yolk or 1% soybean lecithin (L-a-phosphatidyl choline, LAB, product number MC041), 5% glycerol (Chem-Lab NV, Zedelgem, Belgium), 100 IU/mL penicillin, and 100 µg/mL streptomycin. The osmolarity and pH were assessed before adding the cryoprotectants and were adjusted to an osmolarity of 280–300 mOsm (Micro-Osmometer, Loser Type 6, Löser Messtechnik, Berlin, Germany) and a pH value of 7.2–7.3 (PH/mV Temperature Meter, Jenway 3510, Jenway, Staffordshire, UK). The pooled ejaculates were divided into two aliquots of equal volume that were extended using EYE and SBLE. For the extension procedure, each extender was supplemented with different concentrations of BBE—i.e., 0, 250, 500, 750, and 1000 µg/mL—at a semen:extender ratio reaching 1:20 to obtain a final sperm concentration of 150 × 10^6^ sperm/mL. The diluted semen was cooled to 5 °C for 4 h (for equilibration) before being loaded into 0.5 mL straws. The straws were placed 4 cm above liquid nitrogen vapor for 10 min and then immersed in liquid nitrogen. The straws remained in liquid nitrogen until thawing at 37 °C in a water bath for 30 s.

### 2.5. Semen Analysis

Samples were analyzed after the equilibration period (at 5 °C for 4 h), after thawing at 37 °C in a water bath for 30 s, and after incubation of thawed semen at 37 °C and 5% CO_2_ for 2 h.

#### 2.5.1. Progressive Motility

Sperm progressive motility, the ability of sperm to move forward in a long semiarc pattern, was determined by analyzing a drop of diluted semen (10 µL) using a phase-contrast microscope with magnification (100×) supplied with a hot stage at 37 °C (DM 500; Leica, St. Gallen, Switzerland). A 10 µL aliquot of diluted semen was placed on a warm glass slide, covered by a clean coverslip, and examined under the microscope (100× objective). At least 200 spermatozoa from a minimum of four microscopic fields were examined. The progressive motility was estimated on a continuous scale of 0% to 100% [34]. The same professional investigator performed the blind analysis that was conducted in three replicates.

#### 2.5.2. Determination of the Proportion of Alive and Abnormal Sperms

Diluted semen samples were smeared on a glass slide and stained following a dual staining procedure (5% eosin and 10% nigrosin) [35]. Two hundred sperm cells from each sample were examined at a 400× magnification using a light microscope (DM 500; Leica, Switzerland). The number of dead spermatozoa (stained in red) or live spermatozoa (unstained) was counted. Additionally, the morphological abnormalities of the spermatozoa—i.e., spermatozoa bearing head, tail, and cytoplasmic droplet abnormalities—were determined [36].

#### 2.5.3. Plasma Membrane Integrity

A hypo-osmotic swelling test (HOST) solution (75 mOsm/L) was prepared by dissolving 3.67 g of sodium citrate and 6.75 g of fructose in 1000 mL of distilled water. A volume of 50 μL of extended semen was gently mixed with 500 µL of HOST solution and incubated for 30 min at 37 °C. After incubation, a drop of HOST solution containing extended semen was placed on a glass slide, covered with a coverslip, and examined under a phase-contrast microscope (DM 500; Leica, Switzerland) at a 400× magnification. A total of 200 spermatozoa were counted, and the percentage of HOST-positive spermatozoa (having swelled or curled tails) was determined [37].

#### 2.5.4. Acrosome Morphology

A drop of thawed semen was smeared on a pre-warmed slide and air dried. The smears were fixed by immersion in 5% formaldehyde for 30 min and washed under running tap water. Then, the smears were air dried and immersed in buffered Giemsa solution in a Coplin jar for 3 h. Afterward, they were washed under running tap water and dried. The dried smears were analyzed under a phase-contrast microscope at a magnification of 1000× using an oil immersion lens. The percentage of normal acrosomes was calculated for about 200 spermatozoa. The acrosome was considered normal when the staining was clearly and evenly distributed across the spermatozoa area anterior to the equatorial segment [38].

#### 2.5.5. Sperm Apoptosis and Necrosis (Annexin V/Propidium Iodide [PI] Assay)

Annexin V staining of sperm cells and flow cytometry were used as described by Chaveiro, et al. [39]. Briefly, 1 mL of sperm sample was mixed with 2 mL of binding buffer in a 5-mL tube, and 100 µL of sperm suspension was mixed with 5 µL of annexin V (fluorescein isothiocyanate [FITC] labeled, BD Pharmingen™, Cat. No. 51-65874x) and 5 µL of PI (phycoerythrin labeled, BD Pharmingen™, Cat. No. 51-66211E) and incubated for 15 min in the dark at room temperature. After incubation, the samples were suspended in 200 µL of binding buffer. The flow cytometric analysis, (Ex = 488 nm; Em = 350 nm) using the FITC signal detector (FL1) and PI staining by the phycoerythrin emission signal detector (FL2), was conducted on an Accuri C6 Cytometer (BD Biosciences, San Jose, CA, USA), and the data were acquired and analyzed using Accuri C6 software (Becton Dickinson) as described by Masters and Harrison [40]. The percentages of annexin V-positive or -negative (A+ or A−), PI-positive or -negative (PI+ or PI−), and double-positive cells were calculated. Based on these proportions, spermatozoa were classified into four categories as described by Peña, et al. [41]:A− and PI− spermatozoa (no fluorescent signal detected) were classified as viable and recorded as live without plasma membrane dysfunction (live sperm).A+ and PI− spermatozoa were classified as early apoptotic but viable (live sperm).A+ and PI+ spermatozoa were classified as apoptotic with damaged/permeable plasma membranes (dead sperm).A− and PI+ spermatozoa were classified as necrotic as they had completely lost the sperm plasma membrane without signs of apoptosis (dead sperm).

### 2.6. Biochemical Analysis of the Extenders after Thawing

Extenders of frozen–thawed semen were isolated after thawing for all treatment conditions. Extenders were separated from samples by centrifugation for 15 min at 4430× *g* and were stored at −20 °C until analysis. The total antioxidant capacity (TAC), malondialdehyde (MDA) levels, and hydrogen peroxide (H_2_O_2_) levels were measured as published previously [42,43,44]. In brief, the determination of TAC was accomplished by the reaction of antioxidants in the sample with a definite amount of exogenously added hydrogen peroxide. TAC content was measured at a wavelength of 505 nm. The concentration of MDA was measured using thiobarbituric acid. The thiobarbituric acid reacts with malondialdehyde in an acidic medium at a temperature of 95 °C for 30 min to form a thiobarbituric acid reactive product, and the absorbance of the resultant pink product measured 534 nm. H_2_O_2_ reacts with 3,5-dichloro-2-hydroxybenzensulfonic acid and 4-aminophenazone to form a chromophore that is measured at 510 nm. Additionally, the enzymatic activities of aspartate transaminase (AST) and alanine transaminase (ALT) were measured according to the method of [45]. Alkaline phosphatase (ALP) and total acid phosphatase (TAP) activities were determined based on the procedures from [46,47]. All biochemical assays were performed using commercial kits (Biodiagnostic, Egypt) following the manufacturers’ instructions and a spectrophotometer (Spectro UV-VIS Auto UV-2602; Labomed, Los Angeles, CA, USA).

### 2.7. Statistical Analysis

All numerical data have been checked for homogeneity of variance using Lieven’s test and normality of distribution using the Shapiro–Wilk test. The General Linear Model (GLM) procedures in SAS (2004) and two-way analysis of variance (ANOVA) were used for statistical analyses of the data to determine the effects of the extender type, BBE concentration, and their interaction on sperm characteristics and biochemical compositions of the extender. A Duncan’s multiple range test was used to test the differences among the means at *p* < 0.05 [48]. The percentage values were subjected to an arcsine transformation before the ANOVA. Means are presented after being recalculated from the values transformed into percentages. The number of replicates was seven for sperm characteristics, five for biochemical analysis, and three for sperm apoptosis.

## 3. Results

### 3.1. BBE Analysis

The chemical analysis of BBE revealed that TSC levels reached 678.054 µg glucose equivalent/mg extract. The BBE content in TPC was 17.23 mg of gallic acid equivalents (GAE)/g extract and its TFC was 4.98 mg/g extract. The BBE displayed significant antioxidant activity in two widely used assays, the FRAP and DPPH assays (Table 1).

### 3.2. Effects of the Extender Type and BBE Concentration

#### 3.2.1. Sperm Characteristics after Equilibration

The sperm characteristics were assessed in ram semen after equilibration at 5 °C for 4 h. Ram semen diluted with SBLE showed the best sperm characteristics with significantly enhanced vitality and abnormality after the equilibration period compared with the sperm characteristics observed with EYE (Table 2). Sperm vitality was significantly improved in the samples treated with BBE in a concentration-dependent pattern compared with that of the control (Table 2). The SBLE or EYE supplemented with 500, 750, and 1000 µg/mL BBE resulted in the best vitality of ram semen after the equilibration period compared with that of the control (Table 2).

#### 3.2.2. Sperm Characteristics in Thawed Ram Semen

Sperm characteristics were analyzed after thawing ram semen at 37 °C for 30 s. As shown in Table 3, all sperm characteristics were improved in ram semen extended with SBLE, particularly vitality and abnormality, which were significantly better. The addition of BBE at different concentrations to the ram semen diluent significantly improved sperm vitality and membrane integrity in a concentration-dependent manner (Table 3). The addition of BBE to the EYE or SBLE also significantly enhanced the post-thawing progressive motility and vitality of ram semen (Table 3).

#### 3.2.3. Sperm Characteristics in Thawed Ram Semen Incubated at 37 °C and 5% CO_2_ for 2 h

As shown in Table 4, all post-thawed and post-incubation (at 37 °C and 5% CO_2_ for 2 h) sperm characteristics were significantly enhanced in SBLE samples compared with those obtained with EYE. Progressive motility, vitality, and membrane integrity of sperm increased significantly when 500, 750, and 1000 µg/mL of BBE were added (Table 4). The same parameters were further increased after thawing and incubation at 37 °C and 5% CO_2_ for 2 h in the presence of EYE or SBLE supplemented with 500, 750, and 1000 µg/mL of BBE (Table 4).

#### 3.2.4. Antioxidant and Oxidative Markers in the Extender after Thawing Ram Semen

Data in Table 5 show that the antioxidant capacity was significantly increased and MDA and H_2_O_2_ levels were significantly decreased in samples treated with SBLE compared with those in samples incubated with EYE. The same trend was observed with supplementation of the semen extender with BBE, which exerted antioxidant properties in a concentration-dependent pattern (Table 5). SBLE supplementation with concentrations of BBE ranging from 250 to 1000 µg/mL significantly increased the TAC compared with that in controls and in samples incubated with other concentrations of BBE and EYE. In contrast, MDA and H_2_O_2_ levels were significantly decreased in samples treated with different BBE concentrations and SBLE compared with those in the controls and samples exposed to other concentrations of BBE and EYE (Table 5).

#### 3.2.5. Enzymatic Activity in the Extender after the Thawing of Ram Semen

The enzymatic activity of AST and ALT was significantly decreased, whereas ALP and TAP activities were significantly increased in EYE after the thawing of semen compared with the values measured in SBLE (Table 6). Supplementation of ram semen extenders with different concentrations of BBE had no significant impact on these enzymatic activities (Table 6).

#### 3.2.6. Quantification of Viable, Early Apoptotic, Apoptotic, and Necrotic Sperms in Ram Semen after Thawing Using Annexin V/PI Assays

Ram semen extended with SBLE contained significantly earlier apoptotic and apoptotic sperm and significantly fewer necrotic sperm compared with that in semen extended with EYE. In contrast, viable sperm amounts were not affected by the extender type (Table 7). Supplementing semen extenders with BBE significantly increased the percentage of viable sperms and significantly decreased the percentage of early apoptotic and apoptotic sperm in a concentration-dependent pattern (Table 7).

## 4. Discussion

Modification of semen extenders by adding natural bee byproducts for preserving sperm may enhance sperm characteristics, provide energy, and control bacterial contamination or antioxidant activity, thus enabling fertilization [17]. Moreover, several authors have reported that various factors influence the preservability of spermatozoa through cooling or freezing, particularly by affecting the osmotic stress, ice crystal formation, toxicity of the cryoprotectants, and the sensitivity to cryoinjury, which is variable among species [1,11,18,19].

The present study revealed that BBE exhibited antioxidant capacities. DPPH assays measure the reaction between hydrogen atoms and peroxyl radicals in the presence of lipophilic antioxidants [49]. Di Cagno, et al. [50] reported that the TPC and TFC contents of raw BB samples ranged from 317 to 378 mg GAE/100 g of BB and 60 to 272 ± 5 mg quercetin equivalent/100 g of BB, respectively. These findings are consistent with previous studies indicating that BBE is an important source of phenolic compounds with antioxidant activity [24]. Moreover, TPC and TFC generally exhibit higher antioxidant activities and are good antibacterial agents [51]. Tomás et al. [49] reported that the most abundant macronutrient in bee products was carbohydrates, with an average total carbohydrate content of 71 g/100 g in bee byproducts. The carbohydrate contents were up to 72.82% and 58.16 g/100 g of bee byproducts in [25,52], respectively.

The present study showed that sperm characteristics after equilibration, thawing, and thawing followed by incubation were improved by increasing BBE concentrations from 250 to 1000 µg/mL in EYE or SBLE compared with those of extenders free of BBE. Additionally, sperm characteristics during the freezing steps were better with SBLE than those with EYE. Globally, there are mostly reports on the enhancement of sperm parameters induced by adding other bee byproducts—such as propolis, royal jelly, and bee pollen—extracts to semen extenders. However, the effects of enriching extenders with BBE have not been investigated. Nevertheless, Tomás et al. [49] stated that there was no statistically significant difference among the chemical compounds of bee byproducts. Thus, it is clear that bee byproducts—propolis, royal jelly, and bee pollen—allow for the improvement of sperm characteristics during semen preservation. These findings are in agreement with the studies of [15,16] on propolis, [18,19] on royal jelly, and [20,21,22] on pollen grains, confirming the potential of bee byproducts for preserving semen through cooling and freezing. The positive effect of BBE on sperm quality is due to its beneficial unique structural components allowing the preservation of sperm cells as effectively as other bee byproducts. Indeed, Othman, et al. [53] showed that BBE contains carbohydrates, proteins, and fat levels reaching 32.74–59.55, 17.22–18.37, and 21.7–4.80 g/100 g of BBE, respectively. Additionally, other studies [25,52] found high levels of fats, proteins, carbohydrates, and energy sources in BBE samples. These studies emphasized that the positive effects of adding bee byproduct extracts to semen diluents on sperm quality were due to the extracts’ content in minerals and amino acids (for example, L-arginine ranged from 1.96 to 2.66 g/100 g of extract according to Mohammad et al. [25]). Similarly, Bakour et al. [24] found minerals such as calcium, zinc, and phosphorus in BBE. They also reported the presence of other minerals, including sodium, potassium, and magnesium in BBE. Likewise, Adaškevičiūtė, et al. [54] indicated that phosphorus was the most abundant mineral in BBE, followed by potassium and calcium, and other minerals such as magnesium, iron, sodium, manganese, zinc, copper, chrome, cadmium, barium, and lead. Concerning amino acids (such as L-arginine) in BBE, Hegazy, et al. [55] showed that supplying semen dilutions with different amounts of L-arginine improved the properties of semen after thawing. The vitamin content of BBE is greatly related to the improvement in sperm characteristics, vitamins having been shown to enhance sperm functionality [56,57]. Moreover, the better sperm function observed with extenders containing BBE compared with that of BBE-free extenders might be attributed to BBE’s free radicals scavenging and controlling antioxidant power. In this context, Mohammad et al. [25] found that antioxidant substances could remove different types of free radicals such as ROS, including superoxide anion, H_2_O_2_, and hydroxyl radicals and activate the production of enzymatic antioxidants like superoxide dismutase, glutathione peroxidase, and catalase, which improve sperm function. A variety of antioxidants are found in the spermatozoa and seminal plasma, but the antioxidant capacity gradually declines when extending the freezing process; thus, antioxidant supplements are preferentially added to the semen extenders [58]. Consequently, antioxidants are added to semen extenders to minimize either H_2_O_2_ or lipid peroxidation levels and to lessen the decline in semen quality parameters during preservation. The essential amino acid (like L-arginine) content of BBE exerts an antioxidant effect by eliminating the excess of oxygen free radicals [25,55]. Moreover, BBE can limit fatty acid peroxidation, as indicated by the lower levels of MDA [59]. This also reveals that antioxidant properties are attributed to physical and chemical compounds such as TPC and TFC. These mechanisms likely contributed to the favorable effects on antioxidant and oxidative biomarkers after thawing (TAC, MDA, and H_2_O_2_) of EYE or SBLE supplemented with different concentrations of BBE observed in our study. Omar, et al. [60] stated that arginine prevents lipid peroxidation under different peroxidation conditions and it acts as an antioxidant agent by absorbing inactivating superoxide anions, thereby scavenging free radicals. Additionally, minerals and vitamins [61] play an important role in improving antioxidant defenses in semen extenders. To the best of our knowledge, very little information is available about the correlation between the extender type, BBE levels, and enzymatic activities in ram seminal plasma. Hence, the transaminase activities (AST and ALT) in extenders are a good indicator of semen quality because they measure sperm membrane stability [62]. For instance, ALT, AST, and ALP are fundamental for metabolic procedures that provide the energy necessary for sperm motility, viability, and fertility [63]. Thus, the increasing proportion of abnormal spermatozoa in the ejaculate leads to high concentrations of transaminases in the extracellular fluid due to sperm membrane damage and leakage of enzymes from spermatozoa [64]. The significant continuous elevation in the spillage of ALT, AST, and ALP enzymes in the extracellular medium through preservation may be due to the destruction of sperm cell membranes induced by preservation [65]. Here, the supplementation of ram semen extenders with BBE enhanced the proportion of viable sperm and reduced that of apoptotic sperm, suggesting that BBE has anti-programmed death properties. This is in agreement with the results obtained for small ruminants using medicinal plants for semen cryopreservation [38,66,67]. Treatment of high-fat diet-induced obese male rats with bee bread upregulated testicular antioxidant enzymes and downregulated apoptosis genes [68,69].

Our data showed that SBLE generated better results than those obtained with EYE. Similar results were reported using soybean lecithin in rams [15], billy goats [70], and bulls [71].

## 5. Conclusions

SBLE was more effective than EYE for the cryopreservation of ram semen. Additionally, BBE supplementation of extenders improved ram semen quality during the freezing steps in a concentration-dependent pattern. Motility, vitality, and membrane integrity were specifically enhanced in BBE-treated semen. The improvement in these parameters occurred with an amelioration of the antioxidant capacity and enzymatic activity and reduced apoptosis levels compared with those of controls. Further studies are required to determine the effects of BBE on fertility outcomes.

## Figures and Tables

**Table 1 cells-11-03403-t001:** Composition and antioxidant activities (means ± SD) of bee bread extract (BBE).

Sample	TSC µg GE/mg Extract	TPC mg GAE/g Extract	TFC mg/g Extract	DPPH EC_50_ µg/mL	FRAP (µM TE/mg Extract)
BBE	678.54 ± 32.02	17.23 ± 0.71	4.98 ± 0.25	579.6 ± 22.17	26.68 ± 2.01
Trolox (µM) *	-	-	-	24.42 ± 0.87	-

Total soluble carbohydrates (TSC); total phenolic compounds (TPC); total flavonoid compounds (TFC); 2,2-diphenyl-1-picryl-hydrazyl-hydrate (DPPH); ferric-reducing antioxidant power (FRAP); glucose equivalent (GE); gallic acid equivalent (GAE); Trolox equivalent (TE). * Trolox is a positive control.

**Table 2 cells-11-03403-t002:** Effects of the extender type, bee bread extract (BBE) concentration, and their interaction on the proportion (%) of ram sperm characteristics after equilibration at 5 °C for 4 h.

	Progressive Motility	Vitality	Membrane Integrity	Abnormality
**Extender type**				
Egg Yolk (EYE)	76.1 ± 0.65	77.9 ± 0.71 ^b^	77.0 ± 0.95	5.1 ± 0.28 ^a^
Soya been lecithin (SBLE)	78.0 ± 0.72	80.1 ± 0.74 ^a^	78.5 ± 0.81	4.0 ± 0.26 ^b^
*p* value	0.06	0.03	0.24	0.01
**BBE concentration**				
0 µg/mL (Control)	74.3 ± 1.16	76.0 ± 1.17 ^b^	75.3 ± 1.59	4.3 ± 0.44
250 µg/mL	77.1 ± 1.01	78.4 ± 1.06 ^ab^	77.0 ± 1.62	4.5 ± 0.42
500 µg/mL	77.9 ± 1.01	79.6 ± 0.97 ^a^	78.2 ± 1.26	5.0 ± 0.39
750 µg/mL	77.5 ± 0.87	80.0 ± 1.02 ^a^	78.9 ± 1.16	5.1 ± 0.58
1000 µg/mL	78.6 ± 1.22	81.0 ± 1.32 ^a^	79.1 ± 1.25	4.0 ± 0.39
*p* value	0.06	0.02	0.29	0.30
**Extender type × BBE concentration**				
EYE × 0 µg/mL	73.6 ± 1.43	74.6 ± 1.45 ^c^	73.9 ± 2.11	4.6 ± 0.53
EYE × 250 µg/mL	76.4 ± 1.43	77.6 ± 1.70 ^bc^	77.0 ± 2.87	4.9 ± 0.63
EYE × 500 µg/mL	77.1 ± 1.49	79.4 ± 1.15 ^abc^	78.6 ± 2.27	5.6 ± 0.57
EYE × 750 µg/mL	76.4 ± 0.92	78.9 ± 1.06 ^abc^	77.9 ± 1.62	6.3 ± 0.71
EYE × 1000 µg/mL	77.1 ± 1.84	79.1 ± 2.03 ^abc^	77.6 ± 1.67	4.4 ± 0.53
SBLE × 0 µg/mL	75.0 ± 1.89	77.4 ± 1.78 ^bc^	76.7 ± 2.41	4.0 ± 0.72
SBLE × 250 µg/mL	77.9 ± 1.49	79.1 ± 1.32 ^abc^	77.0 ± 1.77	4.1 ± 0.55
SBLE × 500 µg/mL	78.6 ± 1.43	79.7 ± 1.66 ^ab^	77.9 ± 1.32	4.4 ± 0.48
SBLE × 750 µg/mL	78.6 ± 1.43	81.1 ± 1.71 ^ab^	80.0 ± 1.68	4.0 ± 0.72
SBLE × 1000 µg/mL	80.0 ± 1.54	82.9 ± 1.52 ^a^	80.7 ± 1.77	3.6 ± 0.57
*p* value	0.16	0.05	0.54	0.09

^a–c^ within a column, means labeled with different superscripts are significantly different (*p* < 0.05).

**Table 3 cells-11-03403-t003:** Effects of the extender type, bee bread extract (BBE) concentration, and their interaction on the proportion (%) of ram sperm characteristics after thawing.

	Progressive Motility	Vitality	Membrane Integrity	Acrosome Integrity	Abnormality
**Extender type**					
EYE	34.9 ± 0.60	36.6 ± 0.69 ^b^	34.6 ± 0.68	91.8 ± 0.38	9.2 ± 0.32 ^a^
SBLE	36.1 ± 0.62	38.7 ± 0.63 ^a^	35.9 ± 0.80	90.9 ± 0.38	7.7 ± 0.35 ^b^
*p* value	0.08	0.01	0.20	0.12	0.005
**BBE concentration**					
0 µg/mL (Control)	32.1 ± 0.69 ^b^	34.3 ± 0.84 ^c^	32.4 ± 0.85 ^b^	90.8 ± 0.58	8.6 ± 0.70
250 µg/mL	34.3 ± 1.03 ^b^	36.4 ± 1.14 ^bc^	33.7 ± 1.24 ^ab^	91.3 ± 0.59	8.5 ± 0.59
500 µg/mL	36.8 ± 0.85 ^a^	38.8 ± 0.95 ^ab^	36.3 ± 1.01 ^a^	91.6 ± 0.58	8.9 ± 0.46
750 µg/mL	37.1 ± 0.69 ^a^	39.4 ± 0.72 ^a^	36.6 ± 1.26 ^a^	91.1 ± 0.66	8.4 ± 0.52
1000 µg/mL	37.1 ± 0.86 ^a^	39.4 ± 1.03 ^a^	37.1 ± 1.11 ^a^	91.9 ± 0.69	7.9 ± 0.53
*p* value	<0.0001	0.0003	0.02	0.73	0.76
**Extender type × BBE concentration**					
EYE × 0 µg/mL	31.4 ± 0.92 ^c^	32.6 ± 0.84 ^d^	32.1 ± 1.12	90.7 ± 0.89	9.7 ± 0.81
EYE × 250 µg/mL	32.9 ± 1.01 ^bc^	34.7 ± 1.11 ^cd^	32.6 ± 1.13	91.6 ± 0.84	8.9 ± 0.91
EYE × 500 µg/mL	37.1 ± 1.01 ^a^	38.4 ± 1.39 ^abc^	35.7 ± 0.97	92.6 ± 0.78	9.6 ± 0.57
EYE × 750 µg/mL	37.1 ± 1.01 ^a^	39.6 ± 1.21 ^ab^	36.6 ± 1.72	91.3 ± 1.02	9.6 ± 0.57
EYE × 1000 µg/mL	35.7 ± 1.30 ^ab^	37.7 ± 1.57 ^abc^	35.9 ± 1.92	92.7 ± 0.75	8.1 ± 0.67
SBLE × 0 µg/mL	32.9 ± 1.01 ^bc^	36.0 ± 1.18 ^bcd^	32.7 ± 1.36	90.9 ± 0.80	7.6 ± 1.04
SBLE × 250 µg/mL	35.7 ± 1.70 ^ab^	38.0 ± 1.88 ^abc^	34.9 ± 2.22	91.0 ± 0.87	8.1 ± 0.80
SBLE × 500 µg/mL	36.4 ± 1.43 ^ab^	39.1 ± 1.39 ^ab^	36.9 ± 1.83	90.6 ± 0.72	8.1 ± 0.63
SBLE × 750 µg/mL	37.1 ± 1.01 ^a^	39.3 ± 0.89 ^ab^	36.6 ± 1.97	90.9 ± 0.91	7.3 ± 0.64
SBLE × 1000 µg/mL	38.6 ± 0.92 ^a^	41.0 ± 1.09 ^a^	38.4 ± 1.04	91.1 ± 1.14	7.6 ± 0.87
*p* value	0.0004	0.001	0.09	0.68	0.20

^a–d^ Within a column, means labeled with different superscripts are significantly different (*p* < 0.05).

**Table 4 cells-11-03403-t004:** Effects of the extender type, bee bread extract (BBE) concentration, and their interaction on the proportion (%) of sperm characteristics after thawing and incubation of ram semen at 37 °C in 5% CO_2_ for 2 h.

	Progressive Motility	Vitality	Membrane Integrity	Abnormality
**Extender type**				
EYE	29.0 ± 0.67 ^b^	30.5 ± 0.65 ^b^	27.8 ± 0.65 ^b^	11.0 ± 0.33 ^a^
SBLE	31.3 ± 0.78 ^a^	32.9 ± 0.82 ^a^	30.7 ± 0.79 ^a^	9.7 ± 0.34 ^b^
*p* value	0.02	0.02	0.004	0.01
**BBE concentration**				
0 µg/mL (Control)	26.8 ± 1.00 ^c^	28.3 ± 0.92 ^b^	26.2 ± 1.11 ^c^	10.3 ± 0.67
250 µg/mL	28.6 ± 1.10 ^bc^	30.7 ± 1.28 ^ab^	27.9 ± 1.13 ^bc^	10.4 ± 0.58
500 µg/mL	31.1 ± 1.19 ^ab^	32.8 ± 1.28 ^a^	29.6 ± 1.13 ^ab^	10.9 ± 0.51
750 µg/mL	31.8 ± 1.24 ^a^	33.0 ± 1.21 ^a^	31.0 ± 1.24 ^ab^	10.3 ± 0.53
1000 µg/mL	32.5 ± 0.69 ^a^	33.7 ± 0.81 ^a^	31.4 ± 0.95 ^a^	9.9 ± 0.54
*p* value	0.001	0.01	0.01	0.79
**Extender type × BBE concentration**				
EYE × 0 µg/mL	25.0 ± 1.09 ^d^	27.1 ± 0.91 ^c^	24.9 ± 1.58 ^d^	11.4 ± 0.81
EYE × 250 µg/mL	27.1 ± 1.01 ^cd^	29.4 ± 1.34 ^bc^	26.3 ± 1.04 ^cd^	10.7 ± 1.02
EYE × 500 µg/mL	30.7 ± 1.30 ^abc^	31.9 ± 1.39 ^abc^	28.6 ± 1.19 ^abcd^	11.7 ± 0.52
EYE × 750 µg/mL	30.7 ± 1.70 ^abc^	31.7 ± 1.80 ^abc^	29.9 ± 1.70 ^abc^	11.1 ± 0.63
EYE × 1000 µg/mL	31.4 ± 0.92 ^abc^	32.4 ± 1.04 ^ab^	29.3 ± 1.04 ^abcd^	10.0 ± 0.65
SBLE × 0 µg/mL	28.6 ± 1.43 ^bcd^	29.4 ± 1.54 ^bc^	27.6 ± 1.49 ^bcd^	9.1 ± 0.91
SBLE × 250 µg/mL	30.0 ± 1.89 ^abc^	32.0 ± 2.17 ^abc^	29.6 ± 1.89 ^abcd^	10.0 ± 0.62
SBLE × 500 µg/mL	31.4 ± 2.10 ^abc^	33.7 ± 2.20 ^ab^	30.7 ± 1.92 ^abc^	10.0 ± 0.79
SBLE × 750 µg/mL	32.9 ± 1.84 ^ab^	34.3 ± 1.60 ^ab^	32.1 ± 1.82 ^ab^	9.4 ± 0.75
SBLE × 1000 µg/mL	33.6 ± 0.92 ^a^	35.0 ± 1.09 ^a^	33.4 ± 1.17 ^a^	9.7 ± 0.92
*p* value	0.004	0.02	0.01	0.28

^a–d^ Within a column, means labeled with different superscripts are significantly different (*p* < 0.05).

**Table 5 cells-11-03403-t005:** The effects of extender type, bee bread extract (BBE) concentration, and their interaction on antioxidant and oxidative markers in the extender of thawed ram semen.

	TAC (mM/L)	MDA (nmol/mL)	H_2_O_2_ (mM/L)
**Extender type**			
EYE	0.31 ± 0.02 ^b^	21.2 ± 0.75 ^a^	0.082 ± 0.002 ^a^
SBLE	0.58 ± 0.02 ^a^	8.8 ± 0.72^b^	0.045 ± 0.002 ^b^
*p* value	<0.0001	<0.0001	<0.0001
**BBE concentration**			
0 µg/mL (Control)	0.30 ± 0.03 ^c^	20.0 ± 2.69 ^a^	0.074 ± 0.006 ^a^
250 µg/mL	0.45 ± 0.05 ^b^	15.9 ± 2.04 ^b^	0.068 ± 0.007 ^b^
500 µg/mL	0.47 ± 0.05 ^ab^	14.3 ± 2.24 ^bc^	0.063 ± 0.007 ^cb^
750 µg/mL	0.49 ± 0.05 ^ab^	13.3 ± 2.08 ^cd^	0.059 ± 0.007 ^c^
1000 µg/mL	0.52 ± 0.05 ^a^	11.6 ± 1.84 ^d^	0.053 ± 0.006 ^d^
*p* value	<0.0001	<0.0001	<0.0001
**Extender type × BBE concentration**			
EYE × 0 µg/mL	0.21 ± 0.02 ^d^	27.1 ± 0.69 ^a^	0.090 ± 0.003 ^a^
EYE × 250 µg/mL	0.31 ± 0.04 ^c^	21.9 ± 0.88 ^b^	0.088 ± 0.002 ^ab^
EYE × 500 µg/mL	0.32 ± 0.02 ^bc^	20.9 ± 0.94 ^b^	0.082 ± 0.002 ^bc^
EYE × 750 µg/mL	0.35 ± 0.03 ^bc^	19.5 ± 0.85 ^bc^	0.080 ± 0.003 ^c^
EYE × 1000 µg/mL	0.37 ± 0.02 ^bc^	17.0 ± 0.37 ^c^	0.070 ± 0.003 ^d^
SBLE × 0 µg/mL	0.39 ± 0.02 ^b^	12.9 ± 2.66 ^d^	0.058 ± 0.002 ^e^
SBLE × 250 µg/mL	0.59 ± 0.03 ^a^	10.0 ± 0.64 ^de^	0.048 ± 0.002 ^f^
SBLE × 500 µg/mL	0.61 ± 0.02 ^a^	7.8 ± 0.54 ^ef^	0.044 ± 0.002 ^fg^
SBLE × 750 µg/mL	0.64 ± 0.01 ^a^	7.2 ± 0.39 ^ef^	0.038 ± 0.002 ^gh^
SBLE × 1000 µg/mL	0.66 ± 0.02 ^a^	6.2 ± 0.68 ^f^	0.036 ± 0.002 ^h^
*p* value	<0.0001	<0.0001	<0.0001

Total antioxidant capacity (TAC); malondialdehyde (MDA); hydrogen peroxide (H_2_O_2_). ^a–h^ Within a column, means labeled with different superscripts are significantly different (*p* < 0.05).

**Table 6 cells-11-03403-t006:** Effects of the extender type, bee bread extract (BBE) concentration, and their interaction on enzymatic activities in the extender after thawing of ram semen.

	AST (U/mL)	ALT (U/mL)	ALP (IU/L)	TAP (U/L)
**Extender type**				
EYE	42.4 ± 0.93 ^b^	9.4 ± 0.47 ^b^	89.7 ± 2.84 ^a^	8.6 ± 0.20 ^a^
SBLE	49.3 ± 1.18 ^a^	18.2 ± 0.49 ^a^	50.8 ± 2.14 ^b^	6.4 ± 0.19 ^b^
*p* value	<0.0001	<0.0001	<0.0001	<0.0001
**BBE concentration**				
0 µg/mL (Control)	46.4 ± 2.21	13.4 ± 1.87	71.2 ± 7.32	7.3 ± 0.54
250 µg/mL	47.4 ± 1.76	13.0 ± 1.77	70.4 ± 7.92	7.6 ± 0.45
500 µg/mL	45.1 ± 2.17	13.8 ± 1.62	69.7 ± 7.64	7.5 ± 0.42
750 µg/mL	44.8 ± 2.40	14.3 ± 1.37	69.9 ± 7.28	7.6 ± 0.61
1000 µg/mL	45.6 ± 1.64	14.4 ± 1.56	70.1 ± 8.16	7.5 ± 0.42
*p* value	0.83	0.67	1.00	0.98
**Extender type × BBE concentration**				
EYE × 0 µg/mL	41.6 ± 1.47 ^cd^	8.0 ± 0.71 ^b^	89.5 ± 5.29 ^a^	8.4 ± 0.59 ^a^
EYE × 250 µg/mL	43.0 ± 0.89 ^cd^	8.0 ± 1.05 ^b^	90.9 ± 8.21 ^a^	8.6 ± 0.27 ^a^
EYE × 500 µg/mL	43.6 ± 3.33 ^bcd^	9.6 ± 1.12 ^b^	88.7 ± 7.06 ^a^	8.5 ± 0.32 ^a^
EYE × 750 µg/mL	40.6 ± 2.04 ^d^	10.6 ± 0.93 ^b^	88.4 ± 6.96 ^a^	8.9 ± 0.80 ^a^
EYE × 1000 µg/mL	43.2 ± 2.44 ^cd^	10.8 ± 1.07 ^b^	90.9 ± 6.81 ^a^	8.7 ± 0.20 ^a^
SBLE × 0 µg/mL	51.2 ± 2.89 ^ab^	18.8 ± 0.86 ^a^	52.9 ± 6.73 ^b^	6.2 ± 0.58 ^b^
SBLE × 250 µg/mL	51.8 ± 1.85 ^a^	18.0 ± 0.71 ^a^	49.8 ± 1.80 ^b^	6.5 ± 0.52 ^b^
SBLE × 500 µg/mL	46.6 ± 3.01 ^abcd^	18.0 ± 1.34 ^a^	50.7 ± 5.69 ^b^	6.4 ± 0.39 ^b^
SBLE × 750 µg/mL	49.0 ± 3.59 ^abc^	18.0 ± 0.84 ^a^	51.4 ± 4.31 ^b^	6.3 ± 0.41 ^b^
SBLE × 1000 µg/mL	48.0 ± 1.82 ^abcd^	18.0 ± 1.84 ^a^	49.2 ± 5.97 ^b^	6.4 ± 0.31 ^b^
*p* value	0.02	<0.0001	<0.0001	<0.0001

Aspartate transaminase (AST); alanine transaminase (ALT); alkaline phosphatase (ALP); total acid phosphatase (TAP). ^a–d^ Within a column, means labeled with different superscripts are significantly different (*p* < 0.05).

**Table 7 cells-11-03403-t007:** Effects of the extender type, bee bread extract (BBE) concentration, and their interaction on the percentage of viable, early apoptotic, apoptotic, and necrotic sperm in ram semen after thawing using Annexin V/propidium iodide assays.

	Viable (%)	Early Apoptotic (%)	Apoptotic (%)	Necrotic (%)
**Extender type**				
EYE	38.6 ± 2.63	17.7 ± 0.32 ^b^	22.6 ± 2.24 ^b^	21.1 ± 0.47 ^a^
SBLE	39.3 ± 2.30	19.9 ± 0.51 ^a^	25.6 ± 2.16 ^a^	15.1 ± 0.35 ^b^
*p* value	0.14	<0.0001	<0.0001	<0.0001
**BBE concentration**				
0 µg/mL (Control)	24.8 ± 1.06 ^e^	20.2 ± 0.35 ^a^	38.2 ± 0.79 ^a^	16.8 ± 1.73 ^c^
250 µg/mL	34.8 ± 0.38 ^d^	20.1 ± 0.78 ^ab^	25.8 ± 0.64 ^b^	19.3 ± 1.63 ^a^
500 µg/mL	38.0 ± 0.80 ^c^	19.0 ± 1.06 ^b^	24.2 ± 1.07 ^b^	18.9 ± 1.55 ^a^
750 µg/mL	46.5 ± 0.66 ^b^	17.0 ± 0.46 ^c^	18.1 ± 0.64 ^c^	18.4 ± 1.39 ^ab^
1000 µg/mL	50.8 ± 0.68 ^a^	17.8 ± 0.23 ^c^	14.3 ± 0.79 ^d^	17.2 ± 0.58 ^bc^
*p* value	<0.0001	<0.0001	<0.0001	0.0003
**Extender type × BBE concentration**				
EYE × 0 µg/mL	22.9 ± 0.78 ^f^	19.5 ± 0.20 ^bc^	37.1 ± 1.01 ^a^	20.7 ± 0.43 ^ab^
EYE × 250 µg/mL	34.1 ± 0.36 ^d^	18.4 ± 0.36 ^cd^	24.6 ± 0.64 ^bc^	22.9 ± 0.64 ^a^
EYE × 500 µg/mL	39.2 ± 1.07 ^c^	16.7 ± 0.17 ^d^	21.9 ± 0.46 ^cd^	22.2 ± 0.67 ^a^
EYE × 750 µg/mL	45.3 ± 0.55 ^b^	16.5 ± 0.58 ^d^	16.9 ± 0.58 ^ef^	21.4 ± 0.61 ^a^
EYE × 1000 µg/mL	51.6 ± 0.14 ^a^	17.6 ± 0.36 ^cd^	12.6 ± 0.20 ^de^	18.3 ± 0.06 ^bc^
SBLE × 0 µg/mL	26.8 ± 1.04 ^e^	21.0 ± 0.03 ^ab^	39.3 ± 0.92 ^a^	13.0 ± 0.14 ^e^
SBLE × 250 µg/mL	35.5 ± 0.38 ^cd^	21.8 ± 0.12 ^a^	27.1 ± 0.43 ^b^	15.7 ± 0.17 ^d^
SBLE × 500 µg/mL	36.8 ± 0.78 ^cd^	21.3 ± 0.57 ^ab^	26.5 ± 0.52 ^b^	15.5 ± 0.33 ^de^
SBLE × 750 µg/mL	47.7 ± 0.64 ^ab^	17.5 ± 0.67 ^cd^	19.3 ± 0.52 ^g^	15.5 ± 0.77 ^de^
SBLE × 1000 µg/mL	50.0 ± 1.28 ^a^	17.9 ± 0.33 ^cd^	16.0 ± 0.28 ^f^	16.1 ± 0.67 ^cd^
*p* value	<0.0001	<0.0001	<0.0001	<0.0001

^a–g^ Within a column, means labeled with different superscripts are significantly different (*p* < 0.05).

## Data Availability

The data that support the findings of this study are available from the corresponding author, W.A.K., upon reasonable request.

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
