# Peer review of "The Supplementation of Bee Bread Methanolic Extract to Egg Yolk or Soybean Lecithin Extenders Can Improve the Quality of Cryopreserved Ram Semen"

_cells, 2022, doi:10.3390/cells11213403_

Round 1

Reviewer 1 Report

The authors describe the detrimental effects of cryopreservation on spermatozoa and propose to evaluate the effects of supplementing egg yolk extender (EYE) or soybean lecithin extender (SBLE) with bee bread extract (BBE) on the quality of cryopreserved ram semen.

Abstract

- The abstract should be a total of about 200 words maximum and the abstract of the manuscript is 220 words. I recommend that authors adhere to the journal's instructions.

Introduction

The introduction is correctly structured. It highlights the most relevant and current issues in the area of knowledge and ends with the general objective of the study.

Material and methods

Although this section is well structured and described in detail, I have some questions and suggestions for the authors:

- Lines 83-86 how many mg or ug of dry matter did you get after evaporation?

- In which medium was the dry matter of the extract diluted for further analysis?

- Lines 88-95 although the methods described in this section are known, they should be described briefly.

- Lines 160-169 there are other more sensitive and specific fluorescent techniques to assess acrosome integrity, why did the authors use a conventional Giemsa stain? I recommend the authors conduct these analyses using e.g. PNA-FITC or PSA-FITC.

- Lines 170-175 the authors describe the use of Annexin V and PI for apoptosis and necrosis analysis. What is the concentration of annexin V and PI used, and were no post-incubation washes performed?

- Lines 176-179 When flow cytometry is used, it is necessary to give more technical details on its use. Excitation wavelength, filters used for the different fluorophores, etc.

- Lines 193-195 although the methods described in this section are known, they should be described briefly.

- Lines 202-209 the description of the statistical analysis should be detailed. I recommend that the authors indicate how they evaluated Gaussian distribution, the number of replicates, whether the experiments were done in duplicate or triplicate, what is the measure of dispersion used, what is the statistical significance used, etc.

Results

The results are clearly described. I have only one question for the authors:

- Line 251-252, it is not clear which is the control in table 3. 0 ug/mL, EYE x 0 ug/mL or SBLE x 0 ug/mL? If the control is 0 ug/mL, why is there no superscript letter?

Discussion

This section is well developed by the authors. I would only suggest the authors make a more detailed discussion of the anti-programmed death properties of the extract.

Author Response

Reviewer #1:

Thank you so much for your time invested in reading our manuscript, your words of kindness, and your suggestions for improvement.

1

Abstract:

 - The abstract should be a total of about 200 words maximum and the abstract of the manuscript is 220 words. I recommend that authors adhere to the journal's instructions.

R.

Thank you for your comment. The abstract was revised to be 200 words maximum.

2

Introduction:

 The introduction is correctly structured. It highlights the most relevant and current issues in the area of knowledge and ends with the general objective of the study.

R.

Thank you for your comment.

3

Material and methods:

 Although this section is well structured and described in detail, I have some questions and suggestions for the authors:

 - Lines 83-86 how many mg or ug of dry matter did you get after evaporation?

Unfortunately, the quantity produced after evaporation was not calculated, and this will be taken into account in future studies

- In which medium was the dry matter of the extract diluted for further analysis?

1 g of extract was dissolved in 4 ml DMSO.

- Lines 88-95 although the methods described in this section are known, they should be described briefly.

Detailed methods are included in the Supplementary File

- Lines 160-169 there are other more sensitive and specific fluorescent techniques to assess acrosome integrity, why did the authors use a conventional Giemsa stain? I recommend the authors conduct these analyses using e.g. PNA-FITC or PSA-FITC.

 Unfortunately, these analyses are not available in my laboratory, and this will be taken into account in future studies

- Lines 170-175 the authors describe the use of Annexin V and PI for apoptosis and necrosis analysis. What is the concentration of annexin V and PI used, and were no post-incubation washes performed?

The concentration of annexin V and PI used was 5 µL of annexin V (fluorescein isothiocyanate [FITC] labeled, BD Pharmingen™, Cat. No. 51-65874x) and 5 µL of PI (BD Pharmingen™, Cat. No. 51-66211E).

- Lines 176-179 When flow cytometry is used, it is necessary to give more technical details on its use. Excitation wavelength filters used for the different fluorophores, etc.

 Analyze annexin V-FITC binding by flow cytometry (Ex = 488 nm; Em = 350 nm) using FITC signal detector (FL1) and PI staining by the phycoerythrin emission signal detector (FL2).

- Lines 193-195 although the methods described in this section are known, they should be described briefly.

The suggestion has been incorporated. Thank you.

- Lines 202-209 the description of the statistical analysis should be detailed. I recommend that the authors indicate how they evaluated Gaussian distribution, the number of replicates, whether the experiments were done in duplicate or triplicate, what is the measure of dispersion used, what is the statistical significance used, etc.

The suggestion has been incorporated. Thank you.

4

Results 

The results are clearly described. I have only one question for the authors:

- Line 251-252, it is not clear which is the control in table 3. 0 ug/mL, EYE x 0 ug/mL or SBLE x 0 ug/mL? If the control is 0 ug/mL, why is there no superscript letter?

R.

Thanks for your valuable observation. We revised statistical analysis results and added superscript letters.

Discussion:

 This section is well developed by the authors. I would only suggest the authors make a more detailed discussion of the anti-programmed death properties of the extract.

 R

Done

Reviewer 2 Report

-Changing the title from Supplementation of beebread methanolic extract to egg yolk extender or soybean lecithin extenders can improve the quality of cryopreserved ram semen. to “The supplementation of beebread methanolic extract to egg yolk extender or soybean lecithin extender can improve the quality of cryopreserved ram semen

-Grammatical errors can be seen in the text. The whole text should be corrected. Like lines   27, 54, 59, 60, 63,…

-The material and methods have no references. Each section needs a reference.

-Considering the use of different concentrations of BB, in the conclusion and abstract, it should be mentioned whether it is dose-dependent or not and the effective concentration of this compound in improving semen parameters.

-To promote the introduction and discussion from other related references including...

Superior effect of broccoli methanolic extract on control of oxidative damage of sperm cryopreservation and reproductive performance in rats: A comparison with vitamin C and E antioxidant

The effects of broccoli and caraway extracts on serum oxidative markers, testicular structure and function, and sperm quality before and after sperm cryopreservation

Protective effect of nano-vitamin C on infertility due to oxidative stress induced by lead and arsenic in male rats

Author Response

Reviewer #2:

Thank you so much for your time invested in reading our manuscript, your words of kindness, and your suggestions for improvement.

1

Changing the title from Supplementation of beebread methanolic extract to egg yolk extender or soybean lecithin extenders can improve the quality of cryopreserved ram semen. to “The supplementation of beebread methanolic extract to egg yolk extender or soybean lecithin extender can improve the quality of cryopreserved ram semen

R.

Done

2

Grammatical errors can be seen in the text. The whole text should be corrected. Like lines   27, 54, 59, 60, 63,…

R.

The whole manuscript was revised and edited for English by Enago, Crimson Interactive Inc.

3

The material and methods have no references. Each section needs a reference.

R.

The suggestion has been incorporated. Thank you.

4

Considering the use of different concentrations of BB, in the conclusion and abstract, it should be mentioned whether it is dose-dependent or not and the effective concentration of this compound in improving semen parameters.

R.

Thank you for your suggestion. According to your comments, we have modified the abstract and conclusion to mention the improvement by BB was in a concentration-dependent pattern

5

To promote the introduction and discussion from other related references including...

-       Superior effect of broccoli methanolic extract on control of oxidative damage of sperm cryopreservation and reproductive performance in rats: A comparison with vitamin C and E antioxidant

-       The effects of broccoli and caraway extracts on serum oxidative markers, testicular structure and function, and sperm quality before and after sperm cryopreservation

-       Protective effect of nano-vitamin C on infertility due to oxidative stress induced by lead and arsenic in male rats

Thanks for these valuable researches. The suggestion has been incorporated.

Reviewer 3 Report

The aim of the study was to investigate possible effects of beebread methanolic extract supplementation to two different extenders on quality of cryopreserved ram semen.

In general, the main objectives have been achieved and sufficiently supported by experimental data, however, the authors should clarify some methodological procedures.

Minor comments:

-Pag. 3 line 98: the authors should specify how semen quality is tested. Additionally, what is meant by acceptable fertility rates?

-Pag. 3 line 137: Honestly, I am very perplexed as to how the authors assessed progressive motility. Normally the motility parameters of semen were assessed using a computer assisted sperm analysis system and the sample is loaded in a pre-warmed analysis chamber with a depth of 10 μm to allow the mobility of the spermatozoa so that they can be analyzed correctly. Authors should be more specific in describing how the assessment of progressive motility could be done.

- When evaluating the acrosome morphology please provide some pictures to better show what has been described.

- When describing centrifugation please indicate g ad not rpm.

The aim of the study was to investigate possible effects of beebread methanolic extract supplementation to two different extenders on quality of cryopreserved ram semen.

In general, the main objectives have been achieved and sufficiently supported by experimental data, however, the authors should clarify some methodological procedures.

Minor comments

-Pag. 3 line 98: the authors should specify how semen quality is tested. Additionally, what is meant by acceptable fertility rates?

-Pag. 3 line 137: Honestly, I am very perplexed as to how the authors assessed progressive motility. Normally the motility parameters of semen were assessed using a computer assisted sperm analysis system and the sample is loaded in a pre-warmed analysis chamber with a depth of 10 μm to allow the mobility of the spermatozoa so that they can be analyzed correctly. Authors should be more specific in describing how the assessment of progressive motility could be done.

- When evaluating the acrosome morphology please provide some pictures to better show what has been described.

- When describing centrifugation please indicate g ad not rpm.

Author Response

Reviewer 3

The aim of the study was to investigate possible effects of beebread methanolic extract supplementation to two different extenders on quality of cryopreserved ram semen.

In general, the main objectives have been achieved and sufficiently supported by experimental data, however, the authors should clarify some methodological procedures.

  1. We thank the reviewer for the excellent review and constructive comments.

Minor comments:

Q1-Pag. 3 line 98: the authors should specify how semen quality is tested.

  1. Semen quality is tested as we mentioned in the same paragraph (Semen samples were initially evaluated for volume, progressive motility, viability, abnormality, and sperm concentration. Semen samples of volume ≥0.8 mL, progressive motility ≥75%, live sperm ≥85%, abnormal sperm ≤15%, and sperm concentration ≥2.9 ×109/mL).

Q2: Additionally, what is meant by acceptable fertility rates?

  1. We have collected semen from the rams that have exhibited a high conception rate in natural mating during previous seasons. Additionally, these rams were subjected to general and reproductive examination to confirm that they were healthy and free from any health or reproductive diseases.

Q3: Pag. 3 line 137: Honestly, I am very perplexed as to how the authors assessed progressive motility. Normally the motility parameters of semen were assessed using a computer assisted sperm analysis system and the sample is loaded in a pre-warmed analysis chamber with a depth of 10 μm to allow the mobility of the spermatozoa so that they can be analyzed correctly. Authors should be more specific in describing how the assessment of progressive motility could be done.

We are understanding that CASA reduces subjective bias. But, CASA was not in our availability during this study. We have rephrased the progressive motility as follows to be clearer:

A phase-contrast microscope with magnification (100x) and supplied with a hot stage at 37 (DM 500, Leica, Switzerland) was used to determine the percentage of progressive sperm motility which was defined as the ability of sperm to move forward in a long semiarc pattern. A 10 µL aliquot of diluted semen was placed on a warm slide and covered with a coverslip. The same professional investigator performed the blind analysis that was conducted in 3 replicates.

Q4: When evaluating the acrosome morphology please provide some pictures to better show what has been described.

Acrosome morphology was analyzed after thawing at 37 °C in a water bath for 30 s. We used already known procedure and we previously used this protocol and published some photos. Therefore, we did not add photos in this manuscript and we have added a reference from our previous work containing a picture of the parameters of acrosome integrity

Ismail, A. A., Abdel-Khalek, A. K. E., Khalil, W. A., Yousif, A. I., Saadeldin, I. M., Abomughaid, M. M., & El-Harairy, M. A. (2020). Effects of mint, thyme, and curcumin extract nanoformulations on the sperm quality, apoptosis, chromatin decondensation, enzyme activity, and oxidative status of cryopreserved goat semen. Cryobiology, 97, 144-152.

However, if the reviewer think that adding photos will be valuable to our research. We will add it.

Q5: When describing centrifugation please indicate g ad not rpm.

We replaced 6000 rpm with 4430g

Round 2

Reviewer 1 Report

I thank the authors for accepting the suggestions. The manuscript was significantly improved.

Reviewer 2 Report

Corrections have been made.

Reviewer 3 Report

The authors have sufficiently revised the manuscript according to the requirement of reviewer.

The manuscript is suitable for publication in this form.